# An autoinhibited state of 53BP1 revealed by small molecule antagonists and protein engineering

Gaofeng Cui [1,9], Maria Victoria Botuyan [1,9], Pascal Drané[2,9], Qi Hu [1], Benoît Bragantini [1], James R. Thompson [3], David J. Schuller[4], Alexandre Detappe [5], Michael T. Perfetti[6], Lindsey I. James [6,7], Stephen V. Frye [6,7], Dipanjan Chowdhury[2] & Georges Mer [1,8] ✉

The recruitment of 53BP1 to chromatin, mediated by its recognition of histone H4 dimethylated at lysine 20 (H4K20me2), is important for DNA double-strand break repair. Using a series of small molecule antagonists, we demonstrate a conformational equilibrium between an open and a pre-existing lowly populated closed state of 53BP1 in which the H4K20me2 binding surface is buried at the interface between two interacting 53BP1 molecules. In cells, these antagonists inhibit the chromatin recruitment of wild type 53BP1, but do not affect 53BP1 variants unable to access the closed conformation despite preservation of the H4K20me2 binding site. Thus, this inhibition operates by shifting the conformational equilibrium toward the closed state. Our work therefore identifies an auto-associated form of 53BP1—autoinhibited for chromatin binding—that can be stabilized by small molecule ligands encapsulated between two 53BP1 protomers. Such ligands are valuable research tools to study the function of 53BP1 and have the potential to facilitate the development of new drugs for cancer therapy.

The DNA damage response protein 53BP1 (p53-binding protein 1) influences the cell cycle dependency of DNA double-strand break (DSB) repair pathway selection. During G1 phase of the cell cycle, 53BP1 inactivates DSB repair by homologous recombination (HR) and promotes non-homologous end joining (NHEJ). These activities are balanced by the BRCA1-BARD1 E3 ubiquitin ligase, which promotes HR in post-replicative chromatin[1–4], possibly by reshuffling the chromatin localization of 53BP1 during the S and G2 phases[5]. Both 53BP1 and BRCA1-BARD1 recognize DSB-dependent mono-ubiquitylation of histone H2A K15 (H2AK15ub) in chromatin[6–11]. The localization of 53BP1 to chromatin also requires that it binds histone H4 di-methylated at K20

(H4K20me2) via a tandem Tudor domain (53BP1[TT]) (Fig. 1a)[12,13], while BRCA1-BARD1 recruitment requires unmethylated H4K20 recognized by an Ankyrin repeat domain in BARD1[10,11,14]. Our understanding of how 53BP1 and BRCA1-BARD1 contribute to DNA repair pathway selection is incomplete. Investigations of the antagonistic roles of 53BP1 and BRCA1-BARD1 have mostly relied on the identification of separation-of-function mutations. Here, we explore the utility of small molecule ligands that antagonize the recruitment of 53BP1 to chromatin. Blocking the H4K20me2 recognition site of 53BP1[TT] is a prospective means of preventing binding to chromatin. We show that a series of water-soluble small molecules targeting the methyl-lysine binding

[1]Department of Biochemistry and Molecular Biology, Mayo Clinic, Rochester, MN, USA. [2]Department of Radiation Oncology, Dana-Farber Cancer Institute, Boston, MA, USA. [3]The Hormel Institute, University of Minnesota, Austin, MN, USA. [4]Cornell High Energy Synchrotron Source, Cornell University, Ithaca, NY, USA. [5]Institut de Cancérologie Strasbourg Europe, Strasbourg, France. [6]Center for Integrative Chemical Biology and Drug Discovery, Division of Chemical Biology and Medicinal Chemistry, UNC Eshelman School of Pharmacy, University of North Carolina at Chapel Hill, Chapel Hill, NC, USA. [7]Lineberger Comprehensive Cancer Center, University of North Carolina at Chapel Hill School of Medicine, Chapel Hill, NC, USA. [8]Department of Cancer Biology, Mayo Clinic, Rochester, MN, USA. [9]These authors contributed equally: Gaofeng Cui, Maria Victoria Botuyan, Pascal Drané. ✉e-mail: mer.georges@mayo.edu

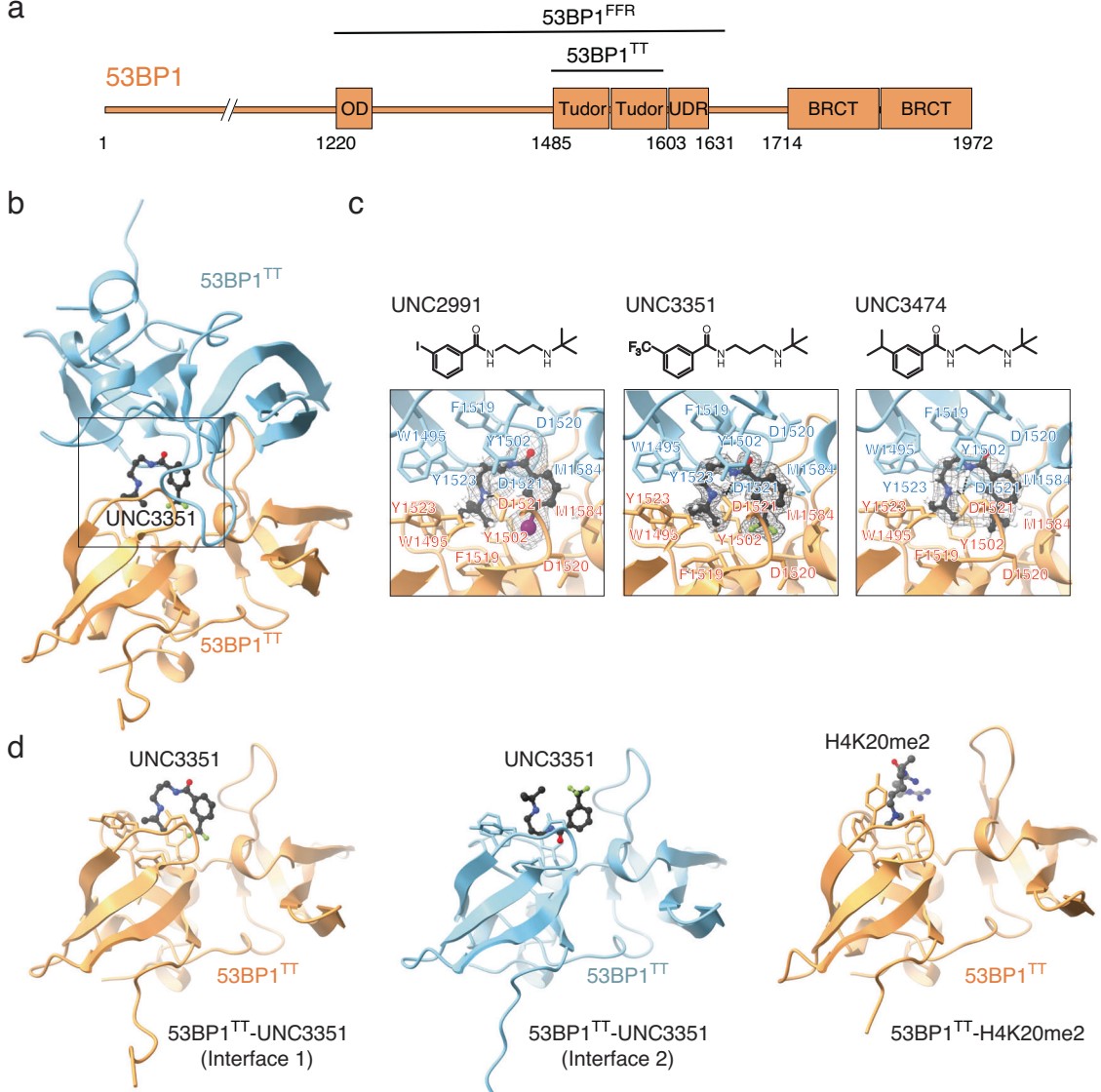

**Fig. 1 | Interaction of 53BP1$^{TT}$ with small molecules probed using X-ray crystallography. a** Domain structure of 53BP1. OD stands for oligomerization domain and UDR for ubiquitin-dependent recognition motif. 53BP1$^{TT}$ is the tandem Tudor domain of 53BP1. 53BP1$^{FFR}$ is the minimal foci-forming region of 53BP1. **b** Crystal structure showing the encapsulation of UNC3351 by two 53BP1$^{TT}$ molecules. **c** Binding interfaces of UNC2991, UNC3351 and UNC3474 with 53BP1$^{TT}$ in the crystal structures of their complexes. The 2mFo-DFc electron density maps contoured at 1σ level are shown as gray mesh around each compound. **d** Binding interfaces in the structure of 53BP1$^{TT}$ bound to UNC3351 (left and middle panels) and to an H4K20me2 peptide[12] (right panel). The side chains in the binding cage of 53BP1$^{TT}$ are shown.

cavity in 53BP1$^{TT}$ function by stabilizing a previously unknown lowly populated autoinhibited homodimeric state of 53BP1$^{TT}$ in which the histone binding surface of H4K20me2 is buried within the protomeric interface. Using these small molecule antagonists and functional 53BP1 constructs consisting of both wild type and mutants engineered to block auto-association via the tandem Tudor domain while retaining chromatin binding capability, we then demonstrate that the autoinhibited state of 53BP1 exists in cells as well. This work paves the way for further exploration of 53BP1 function and modes of action and provides insight for the design of more potent and selective 53BP1 antagonists.

## Results

### A 53BP1 tandem Tudor homodimer encapsulates small molecules in solution

We previously identified a fragment-like small molecule, UNC2170, that binds the aromatic methyl-lysine binding cage of 53BP1$^{TT}$ with a

dissociation constant ($K_d$) of $22 \pm 2.5$ μM and is competitive with an H4K20me2 peptide[15]. Intriguingly, in a crystal structure of UNC2170 bound to 53BP1$^{TT}$, the small molecule ligand was in contact with two 53BP1$^{TT}$ molecules. Our structural characterization of 53BP1$^{TT}$ in complex with the related compounds UNC2991 ($K_d = 3.9 \pm 0.4$ μM), UNC3351 ($K_d = 6.8 \pm 0.4$ μM) and UNC3474 ($K_d = 1.0 \pm 0.3$ μM) (Supplementary Fig. 1a) revealed a similar three-dimensional (3D) arrangement wherein each ligand not only blocked the methyl-lysine binding site but was encapsulated in a cavity generated by two 53BP1$^{TT}$ molecules (Fig. 1b, c and Table 1). This consistent burial of the histone binding surface of 53BP1$^{TT}$ by dimerization was unexpected. While crystallization could drive 53BP1$^{TT}$ homodimerization as has been shown for several other systems[16], that all four 3D structures were comparable despite differing crystallization conditions and space groups suggested that some 53BP1$^{TT}$ dimer may also exist in solution, even though all previous NMR spectroscopy studies only reported a monomeric state for 53BP1$^{TT}$ (Fig. 1d)[12,13,17–19]. Since the

**Table 1 | X-ray data collection and refinement statistics**

| | 53BP1$^{TT}$-UNC2991 (PDB 6MXX) | 53BP1$^{TT}$-UNC3351 (PDB 6MXY) | 53BP1$^{TT}$-UNC3474 (PDB 6MXZ) | 53BP1$^{TT}$-PN (PDB 6MY0) | 53BP1$^{TT}$-CC (PDB 8U4U) |
|---|---|---|---|---|---|
| **Data collection** | | | | | |
| Space group | $P2_1 2_1 2_1$ | $P3_1 21$ | $P2_1 2_1 2_1$ | $C2 2 2_1$ | $P2_1 2_1 2_1$ |
| Cell dimensions | | | | | |
| *a, b, c* (Å) | 68.94, 159.43, 181.19 | 60.57, 60,57, 138.22 | 68.67, 160.71, 182.65 | 73.10,141.27, 47.06 | 72.50, 161.83, 179.95 |
| *α, β, γ* (°) | 90, 90, 90 | 90, 90, 120 | 90, 90, 90 | 90, 90, 90 | 90, 90, 90 |
| Resolution (Å) | 50–2.30 (2.38–2.30)$^a$ | 50–1.62 (1.68–1.62) | 50–2.50 (2.59-2.50) | 50–2.20 (2.28–2.20) | 50–3.79 (3.93–3.79) |
| $R_{merge}$ | 0.077 (1.06) | 0.074 (0.741) | 0.045 (0.436) | 0.087 (0.553) | 0.120 (0.614) |
| $I / \sigma I$ | 21.3 (1.3) | 30.6 (2.7) | 20.5 (2.4) | 23.4 (5.3) | 15.7 (3.2) |
| Completeness (%) | 97.5 (75.3) | 99.7 (98.0) | 97.2 (76.4) | 98.6 (87.4) | 89.4 (76.8) |
| Redundancy | 14.0 (6.7) | 11.2 (8.0) | 6.0 (6.1) | 12.2 (11.4) | 5.7 (5.0) |
| **Refinement** | | | | | |
| Resolution (Å) | 2.30 | 1.62 | 2.50 | 2.20 | 3.79 |
| No. of unique reflections | 87,459 | 37,781 | 68,884 | 12,596 | 19,457 |
| $R_{work} / R_{free}$ | 0.206 / 0.232 | 0.179 / 0.213 | 0.191 / 0.223 | 0.187 / 0.233 | 0.246 / 0.300 |
| No. atoms | | | | | |
| Protein | 10,054 | 2004 | 10,562 | 1887 | 9490 |
| Ligand/ion | 302 | 46 | 221 | N/A | N/A |
| Water | 411 | 348 | 667 | 129 | N/A |
| *B*-factors | | | | | |
| Protein | 55.90 | 19.77 | 56.25 | 27.71 | 142.67 |
| Ligand/ion | 124.93 | 43.04 | 49.13 | N/A | N/A |
| Water | 48.00 | 31.33 | 49.02 | 31.13 | N/A |
| R.m.s. deviations | | | | | |
| Bond lengths (Å) | 0.004 | 0.019 | 0.003 | 0.004 | 0.002 |
| Bond angles (°) | 0.93 | 1.71 | 0.93 | 0.97 | 0.40 |

$^a$Values in parentheses are for highest-resolution shell. All data were collected from single crystals.

homodimerization interface in the crystal structures buries the histone binding surface of 53BP1$^{TT}$ (Fig. 1c, d), such a conformation would be autoinhibitory for chromatin binding. Small molecules that stabilize this autoinhibited state of 53BP1$^{TT}$—including in the context of full-length 53BP1—could potentially inhibit 53BP1 recruitment to DSBs.

Using analytical ultracentrifugation (AUC) sedimentation velocity, we showed that the addition of UNC3474 to 53BP1$^{TT}$ caused a change in sedimentation coefficient that was consistent with the dimerization of 53BP1$^{TT}$ (Fig. 2a). Moreover, changes in small-angle X-ray scattering (SAXS)[20] data upon 53BP1$^{TT}$-UNC3474 complex formation agreed well with 53BP1$^{TT}$ homodimerization, including an increase in the radius of gyration (Supplementary Fig. 1b and Supplementary Table 1). As a control, we used the small molecule UNC1118, which interacts with 53BP1$^{TT}$ via a dimethyl-lysine-mimicking pyrrolidine-piperidine motif (IC$_{50}$ = 7.4 ± 1.1 μM) as we previously showed[21]. We confirmed this interaction by measuring a $K_d$ of 8.5 ± 0.9 μM (Supplementary Fig. 1c). We hypothesized that UNC1118 would be too large to be buried within a 53BP1$^{TT}$ homodimer. The addition of UNC1118 produced SAXS data similar to those of 53BP1$^{TT}$ alone, indicating that UNC1118 is not capable of binding the homodimer (Supplementary Fig. 1b). Ab initio derivation of molecular envelopes from the SAXS scattering curves of 53BP1$^{TT}$ and 53BP1$^{TT}$-UNC3474 disclosed changes consistent with 53BP1$^{TT}$ homodimerization in the presence of UNC3474 (Supplementary Fig. 1b).

Investigation by solution NMR spectroscopy also revealed the dimeric conformation first identified in our 53BP1$^{TT}$ crystal structures. Titration of UNC2170, UNC2991, UNC3351 and UNC3474 into $^{15}$N-labeled 53BP1$^{TT}$ led to changes in several signal positions in the $^1$H–$^{15}$N heteronuclear single quantum coherence (HSQC) NMR spectrum of 53BP1$^{TT}$ indicative of slow exchange on the chemical shift time scale (Supplementary Fig. 2a). These chemical shift perturbations were more extensive than could be expected for just the small molecule binding surface and also mapped to amino acids located at the 53BP1$^{TT}$ dimer interface in our crystal structures (Supplementary Fig. 2b). Several of the 53BP1$^{TT}$ residues that are close to the small molecule binding site had two different NMR signals. This is consistent with loss of symmetry observed in our crystal structures, allowing interactions by each of the two 53BP1$^{TT}$ protomers with a different "side" of the bound small molecule (Fig. 1c, d, Fig. 2b, Supplementary Fig. 2b). NMR relaxation data further supported an increase in molecular weight consistent with 53BP1$^{TT}$ dimerization in the presence of UNC3474 as can be seen from an increase in R$_2$ relaxation rates for the 53BP1$^{TT}$-UNC3474 complex (Supplementary Fig. 3a). In comparison, binding of UNC1118 to 53BP1$^{TT}$ had no marked effect on the relaxation rates of 53BP1$^{TT}$ (Supplementary Fig. 3a).

In addition, for the lowest affinity compound, UNC2170, a slow exchange could be quantified between each of the two protomers in the 53BP1$^{TT}$ dimer with interconversion rates of 1.7 s$^{-1}$ and 1.2 s$^{-1}$ at 25 °C using 2D longitudinal $^1$H-$^{15}$N-heteronuclear ZZ-exchange NMR spectroscopy[22–24] (Fig. 2c). This exchange likely reflects the interconversion between the two 53BP1$^{TT}$ protomers binding the two different "sides" of the ligand. In the presence of the other compounds, which have higher affinity than UNC2170 for 53BP1$^{TT}$, the exchange between the two 53BP1$^{TT}$ protomers was too slow to be quantifiable under the same conditions. Taken together, our data demonstrate that like in the crystal structures, the small molecules are buried at the interface of a 53BP1$^{TT}$ homodimer in solution.

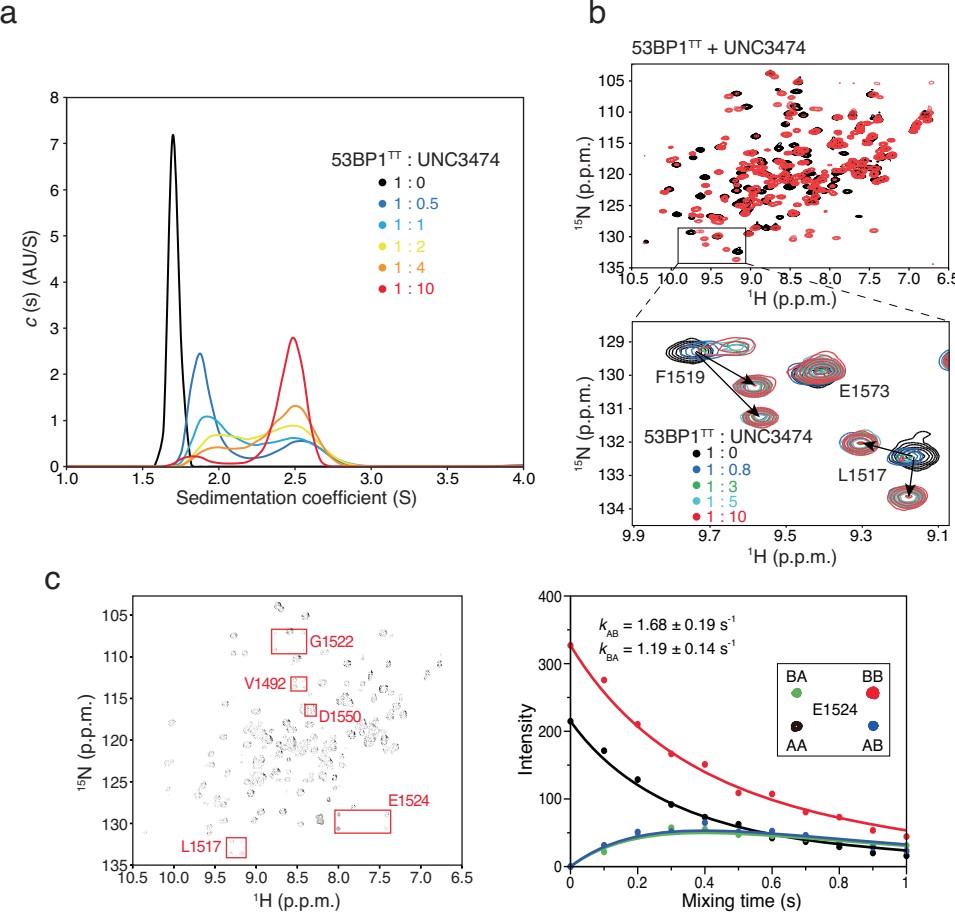

**Fig. 2 | Oligomerization of 53BP1$^{TT}$-UNC3474 probed using analytical ultracentrifugation (AUC) and NMR spectroscopy. a** Sedimentation velocity analysis of 53BP1$^{TT}$ (20 μM) at different concentrations of UNC3474 using AUC. The sedimentation coefficient distributions are shown for free 53BP1$^{TT}$ (black) and for 53BP1$^{TT}$ at increasing concentrations of UNC3474; 10 μM (blue), 20 μM (light blue), 40 μM (yellow), 80 μM (orange) and 200 μM (red). **b** Top: Overlay of $^1$H-$^{15}$N HSQC NMR spectra of $^{15}$N-labeled 53BP1$^{TT}$, free (black) and bound to UNC3474 (red).

Bottom: Examples of doubling of resonances (L1517 and F1519 signals) at different 53BP1$^{TT}$:UNC3474 molar ratios. **c** $^1$H-$^{15}$N-heteronuclear ZZ-exchange NMR spectroscopy of the interaction of 53BP1$^{TT}$ with UNC2170 at 25 °C using 1 mM $^{15}$N-labeled 53BP1$^{TT}$ and 9 mM UNC2170. Fitting was done for the signals of E1524 (inset). The black and red curves show the decay of the auto peaks (AA and BB) and blue and green curves show the buildup and decay of the exchange peaks (AB and BA). The interconversion rates are indicated.

## Identification of a weakly populated homodimeric state of 53BP1 tandem Tudor domain in vitro

The small molecules may cause 53BP1$^{TT}$ to homodimerize or they could stabilize a weakly populated, pre-existing 53BP1$^{TT}$ homodimer that evaded prior detection. This distinction is important. If auto-association, which buries the H4K20me2 binding surface in 53BP1$^{TT}$, also occurs in the context of full-length 53BP1, then 53BP1 may employ autoinhibition as a regulatory mechanism for chromatin recruitment.

In our crystal structures (Fig. 1b, c), W1495 of 53BP1$^{TT}$ is essential for small molecule binding and it also contributes to the 53BP1$^{TT}$ homodimer interface. Several other inter-53BP1$^{TT}$ contacts in the crystal structures, however, are distant from the ligand binding site and might be exploited to disrupt dimer formation while not affecting binding of the small molecules. Such inter-53BP1$^{TT}$ contacts include the hydrophobic interactions between Y1552 and L1518, and salt bridges involving the E1549-R1583, D1550-R1490, and D1550-K1505 pairs (Fig. 3a). To test the possibility of a pre-existing 53BP1$^{TT}$ homodimer, we created a 53BP1$^{TT}$ mutant (53BP1$^{TT}$-PN) in which E1549 and D1550 were replaced by a proline and an asparagine, respectively, to introduce a 3D structure-preserving β-turn in 53BP1$^{TT}$. These mutations eliminate the six salt bridges at the 53BP1$^{TT}$ dimer interface as mentioned above but do not affect the small molecule-binding surface. We verified that the structural integrity of 53BP1$^{TT}$-PN was preserved by

determining a 2.20 Å resolution crystal structure of this protein variant (Fig. 3b, Table 1). The backbone dynamic properties of wild-type 53BP1$^{TT}$ and 53BP1$^{TT}$-PN were quite similar as shown using NMR relaxation measurements (Supplementary Fig. 3b). Like wild-type 53BP1$^{TT}$, 53BP1$^{TT}$-PN bound dimethylated histone H4 (H4K$_C$20me2) and dimethylated p53 (p53K382me2) peptides as expected[12,13,18,19] (Fig. 3c), but unlike WT 53BP1$^{TT}$, 53BP1$^{TT}$-PN did not interact with UNC2170, UNC2991 or UNC3474 (Fig. 3d). There were no changes in the $^1$H-$^{15}$N HSQC spectra of $^{15}$N-labeled 53BP1$^{TT}$-PN after addition of up to fourfold molar excess of each of these small molecules (Fig. 3d). As a control, we also examined the interaction of 53BP1$^{TT}$-PN with the compound UNC1078, which like UNC1118, is a larger molecule that interacts with 53BP1$^{TT}$ as a monomer via an established dimethyl-lysine-mimicking pyrrolidine-piperidine motif and does not cause dimerization (Supplementary Fig. 4a). As expected, 53BP1$^{TT}$-PN also bound UNC1078 (Fig. 3c). The most likely interpretation for the absence of interaction of 53BP1$^{TT}$-PN with UNC2170, UNC2991 and UNC3474 is that these small molecules do not cause 53BP1$^{TT}$ dimerization directly. Instead, they only bind to and stabilize a pre-existing 53BP1$^{TT}$ homodimer population, shifting the equilibrium toward homodimer. When overlaid, the $^1$H-$^{15}$N HSQC spectra of wild-type 53BP1$^{TT}$ and 53BP1$^{TT}$-PN showed small chemical shift changes that map to the inter-53BP1$^{TT}$ interface in 53BP1$^{TT}$-UNC3474. This is further evidence for a preexisting

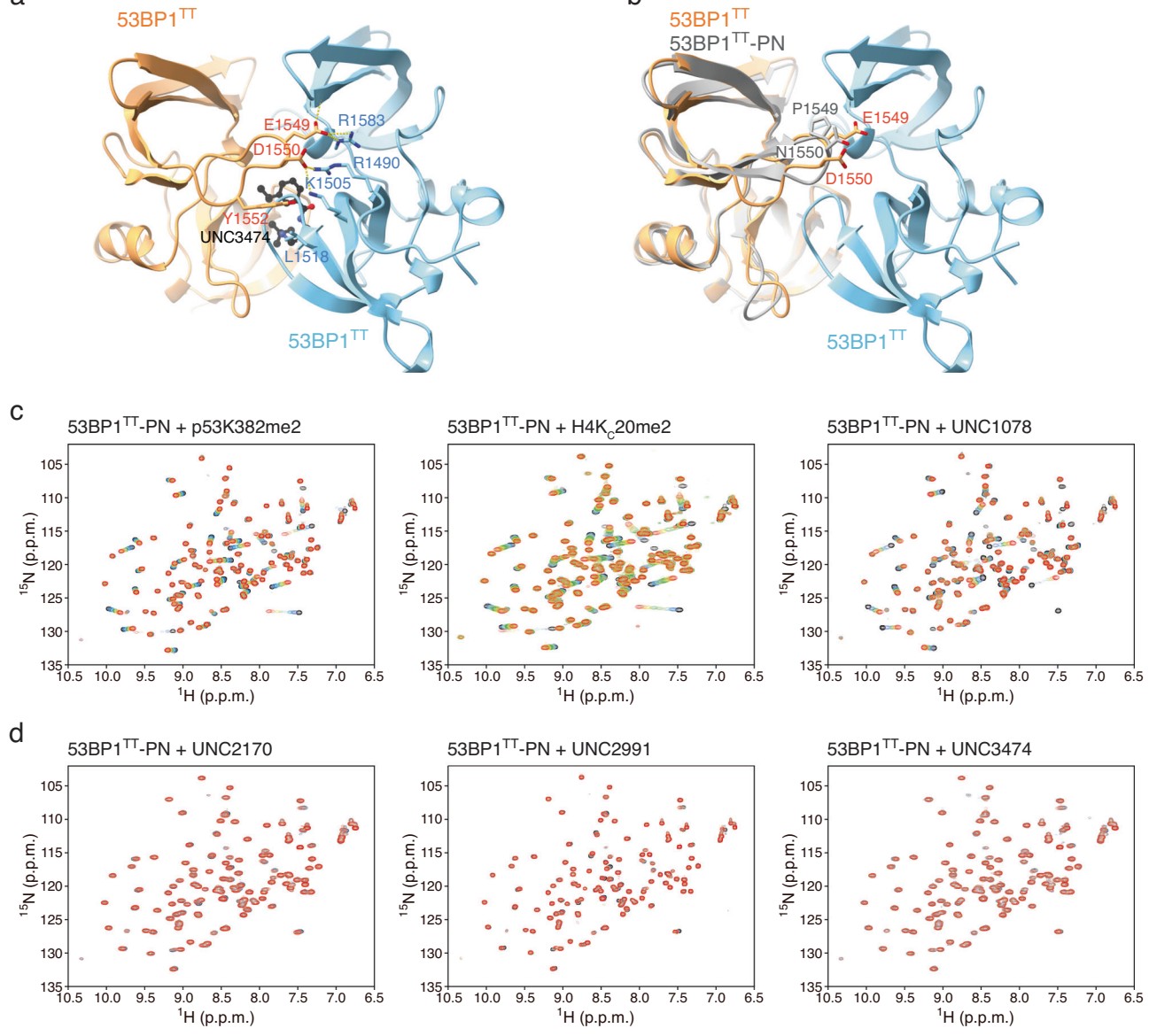

**Fig. 3 | Small molecules stabilize a pre-existing lowly populated 53BP1$^{TT}$ homodimer. a** Key inter-53BP1$^{TT}$ contacts in the structure of 53BP1$^{TT}$-UNC3474 complex. Hydrogen bonds and salt bridges are represented by yellow dashed lines. **b** Structural overlay of wild-type 53BP1$^{TT}$ and mutant 53BP1$^{TT}$-PN in which E1549 and D1550 are replaced by a proline and an asparagine, respectively. **c** NMR spectroscopy-monitored titration of $^{15}$N-labeled 53BP1$^{TT}$-PN with non-labeled

H4K$_C$20me2 and p53K382me2 peptides and small molecule UNC1078. Shown in different colors are the overlaid $^{1}$H-$^{15}$N HSQC spectra of 53BP1$^{TT}$-PN recorded without (black) and with increasing amounts of added compounds, up to fourfold molar excess (red). **d** Same as **c** but for the titration of 53BP1$^{TT}$-PN with small molecules UNC2170, UNC2991, and UNC3474.

small population of 53BP1$^{TT}$ homodimer that is eliminated in 53BP1$^{TT}$-PN (Supplementary Fig. 4b).

As an added test of this binding mechanism, we designed a disulfide-crosslinked[25] 53BP1$^{TT}$ variant that assembles into a stable dimer in the absence of a ligand. Using the software Disulfide by Design[26] applied to the highest resolution structure in Table 1 (Protein Data Bank entry: 6MXY), we selected E1549 and E1567 for replacement by cysteines. These mutations readily linked two 53BP1$^{TT}$ molecules covalently into a homodimer (53BP1$^{TT}$-CC) via formation of two intermolecular C1549-C1567 disulfide bridges as shown using AUC sedimentation velocity (Fig. 4a). A low resolution (3.79 Å) model of 53BP1$^{TT}$-CC derived from X-ray crystallography data was sufficient to demonstrate that both protomers were arranged similarly as wild-type 53BP1$^{TT}$ bound to our small molecules with a root-mean-square deviation for backbone atoms of 0.94 Å. (Fig. 4b, Table 1). Using

NMR spectroscopy, we independently verified that 53BP1$^{TT}$-CC mimicked the homodimeric state of 53BP1$^{TT}$ bound to UNC3474 (Fig. 4c). Noticeably, the addition of UNC3474 led to the splitting of several NMR signals in the $^{1}$H-$^{15}$N HSQC spectra of 53BP1$^{TT}$-CC. The addition of the reducing agent dithiothreitol (DTT) changed 53BP1$^{TT}$-CC to a monomeric state as expected, which, like 53BP1$^{TT}$-PN, had no affinity for UNC3474 (Fig. 4c). We therefore created 53BP1$^{TT}$ variants that can be switched on and off in vitro for ligand binding. These mutations may be useful for the study of a longer, functional 53BP1 in cells.

### Demonstration of an autoinhibited state of 53BP1 in cells using a small molecule ligand

After showing that an autoinhibited state of 53BP1 exists in vitro and can be stabilized by small molecules, we next evaluated the effect of

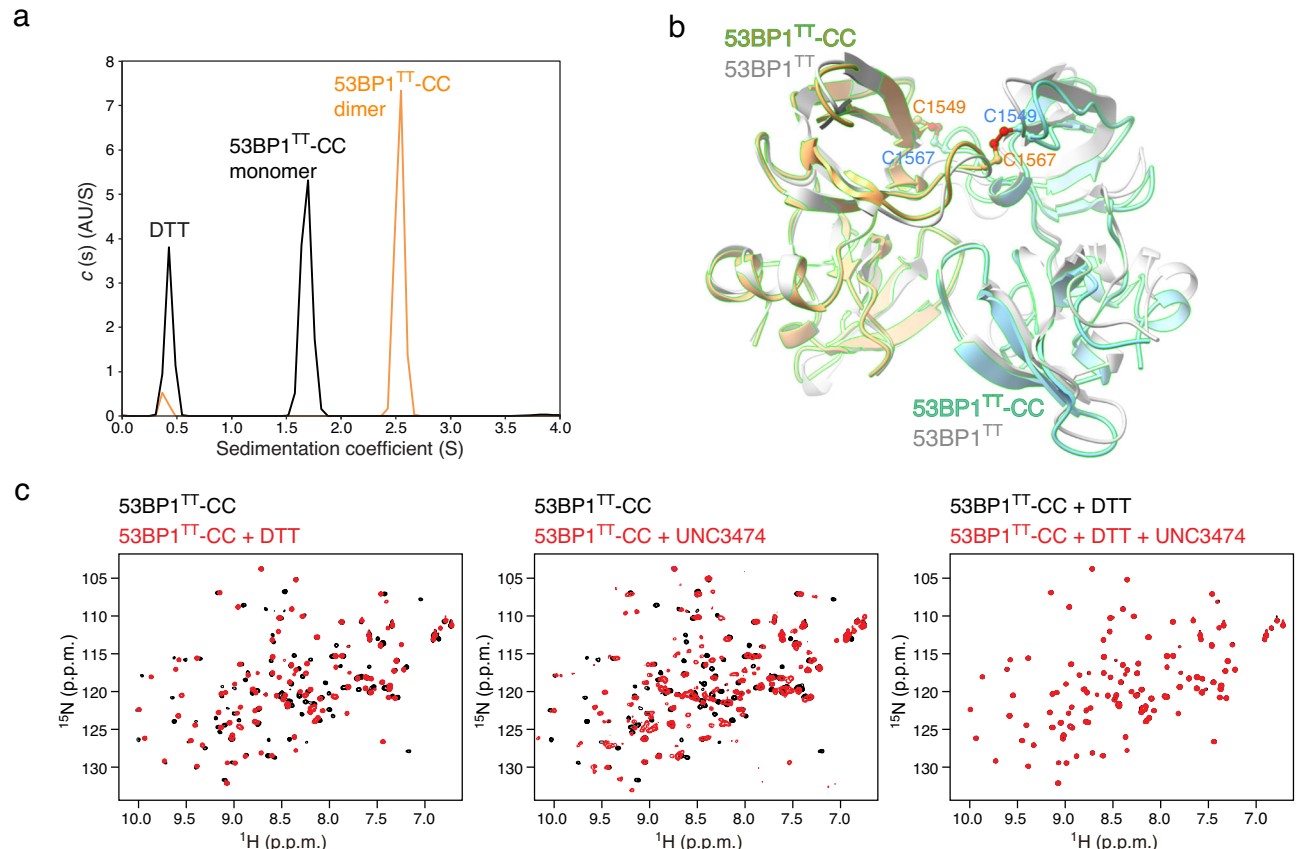

**Fig. 4 | Engineering of a disulfide-crosslinked 53BP1[TT] homodimer that binds UNC3474. a** Sedimentation velocity analysis of 53BP1[TT]-CC (20 µM) in the presence and absence of 10 mM DTT. **b** Crystal structure of 53BP1[TT]-CC (blue and orange) showing the two disulfide bridges overlaid to the crystal structure of 53BP1[TT]-

UNC3474 (UNC3474 omitted) in gray. **c** Effects of adding 2 mM DTT and UNC3474 on the [1]H-[15]N HSQC spectrum of [15]N-labeled 53BP1[TT]-CC (1:1 UNC3474:53BP1[TT]-CC molar ratio).

small molecule UNC3474 on 53BP1 localization to chromatin in cells in response to DNA damage. We first reconstituted mammalian U2OS cell lines to stably express a minimal segment of 53BP1 protein—the oligomeric foci-forming region (53BP1[FFR]: residues 1220–1711) (Fig. 1a)— that contains the tandem Tudor domain and is necessary for DNA damage site recruitment[7,27–29] (Fig. 5a). Cells were generated that express wild-type (WT) 53BP1[FFR] and the mutants 53BP1[FFR]-PN and 53BP1[FFR]-CC.

Using immunofluorescence microscopy, we observed robust formation of ionizing radiation-induced foci (IRIF) after cell exposure to 1 Gy of X-ray radiation for WT 53BP1[FFR], 53BP1[FFR]-PN and 53BP1[FFR]-CC (Fig. 5b). Upon treatment with different concentrations of UNC3474, IRIF formation was inhibited in a dose-dependent manner for WT 53BP1[FFR] (Fig. 5b, c). Under identical experimental conditions, there was no significant change in 53BP1[FFR]-PN and 53BP1[FFR]-CC IRIF upon treatment with UNC3474 (Fig. 5c). The Ser139-phosphorylated histone H2A.X (γH2A.X) IRIF served as a marker for DNA damage in these experiments (Fig. 5d). As we showed in vitro, 53BP1[TT]-PN and reduced-state 53BP1[TT]-CC cannot dimerize and are thus insensitive to UNC3474 and to the related, lower affinity compounds UNC2170 and UNC2991 (Fig. 3d). 53BP1[FFR]-CC is expected to be in a reduced state in mammalian cells owing to the reductive nuclear environment[30,31] and activation of antioxidant enzymes in response to ionizing radiation[32]. 53BP1[FFR]-CC would therefore be predicted to behave like 53BP1[FFR]-PN. These results show that UNC3474 inhibits the recruitment of 53BP1 to DSBs by stabilizing a pre-existing autoinhibited state of 53BP1 in cells. We have therefore demonstrated the existence of an autoinhibited form of 53BP1 with a small molecule ligand used as a chemical probe.

## Discussion

Numerous proteins harbor a methyl-lysine binding cage like that of 53BP1[33], making the development of ligands that are selective for a single methyl-lysine reader protein challenging[34]. Nevertheless, a few selective ligands for methyl-lysine readers have been developed[35]. For example, UNC1215 is a bivalent chemical probe that potently binds chromatin-interacting transcriptional repressor L3MBTL3 by inducing the formation of a 2:2 protein:ligand complex[36]. In a radically different mechanism, UNC3474 and the structurally related small molecules we characterized here stabilize a pre-existing auto-associated form of 53BP1[TT] with a 2:1 protein:ligand stoichiometry. Our studies revealed that these compounds bind the aromatic cage of 53BP1[TT], but only when 53BP1[TT] homodimers exist. We showed that 53BP1[TT] constructs engineered to prevent auto-association—while preserving the ligand binding cage—have no affinity for the small molecule ligands, and yet, they bind H4K[C]20me2 and p53K382me2 like wild-type 53BP1[TT].

Using these small molecules as chemical probes, we demonstrated that a pre-existing autoinhibited form of 53BP1 exists in cells. The function of autoinhibited 53BP1 is not yet understood. Nevertheless, we can speculate that this state may be stabilized in vivo under certain conditions to regulate the activity of 53BP1. We note that full-length 53BP1 and 53BP1[FFR] are constitutive homomultimers[37] via an oligomerization domain of unknown structure located upstream from the tandem Tudor domain (Fig. 1a)[38–41]. In the context of oligomeric 53BP1[FFR], the auto-association of 53BP1[TT] and its stabilization by small molecule ligands reported here are likely amplified by the avidity effect, providing a unique and highly specific means of inhibiting the interaction of 53BP1 with chromatin. In future studies, we will explore

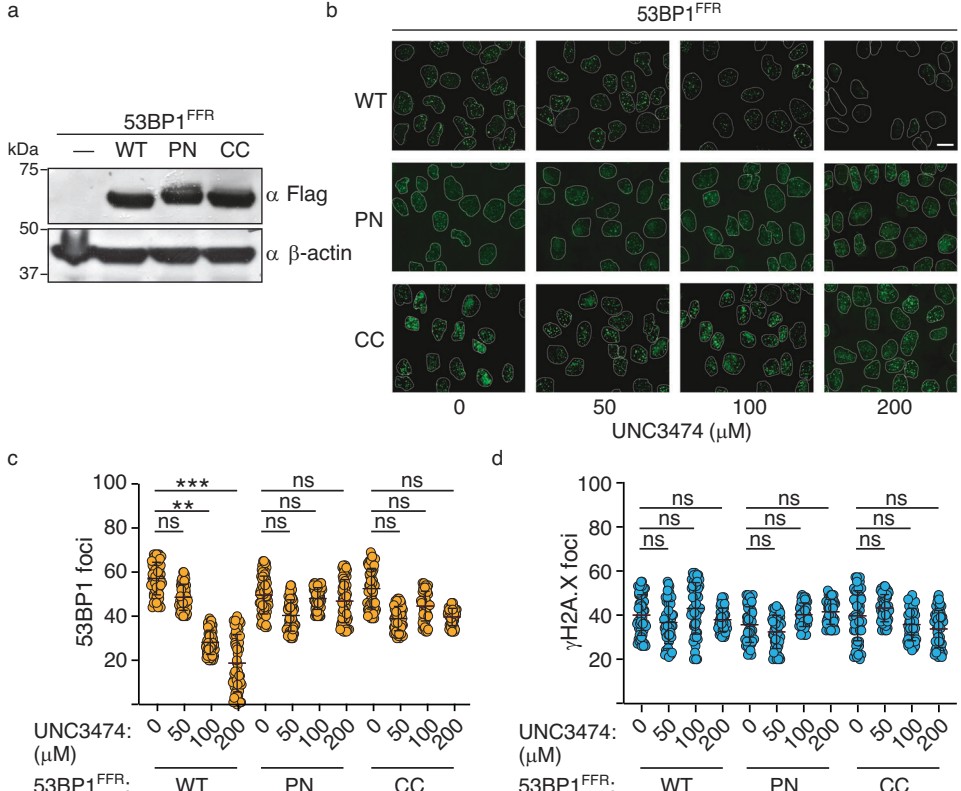

**Fig. 5 | Demonstration of an autoinhibited state for 53BP1 chromatin recruitment in cells using UNC3474 as a chemical probe. a** Immunoblotting showing the expression of the indicated 53BP1$^{FFR}$ construct in stably transfected U2OS cells. β-actin staining was used as control. Source data are provided as a Source Data file. **b** Representative data for the effect of UNC3474 on WT 53BP1$^{FFR}$, 53BP1$^{FFR}$ -PN and 53BP1$^{FFR}$-CC IRIF formation after cell exposure to 1 Gy of X-ray radiation. The scale bar represents 10 μm. The experiments were repeated three times independently with similar results. **c** Effect of UNC3474 on WT 53BP1$^{FFR}$, 53BP1$^{FFR}$-PN and 53BP1$^{FFR}$-

CC IRIF number. IRIF from more than 100 cells were counted in each condition. Data represent mean values ± s.d. A representative plot of $n = 3$ independent experiments is shown. $P$ values were calculated using a two-tailed $t$-test. For (**c**) and (**d**), the labels *, ** and *** indicate $P < 0.05$, $P < 0.01$ and $P < 0.001$, respectively, and ns means not significant. Source data are provided as a Source Data file. **d** Number of γH2A.X IRIF for the cells analyzed in (**c**). A representative plot of $n = 3$ independent experiments is shown. Source data are provided as a Source Data file.

the influence of these ligands on other functions of 53BP1 reliant upon its interactions with the regulatory protein TIRR[28,40,42–44] and K382me2-carrying p53 tumor suppressor[18,45]. Ligand-binding to 53BP1 is predicted to disrupt these interactions.

Our work is expected to facilitate the structure-based design of improved ligands that are more potent in stabilizing the 53BP1$^{TT}$ homodimer and in inhibiting 53BP1 in cells. Such molecules would help further dissect aspects of 53BP1 modes of action. These molecules could also be used to promote DNA repair by homologous recombination and thereby increase the effectiveness of genome editing using CRISPR-Cas9 or related systems[46,47]. Furthermore, pharmacological inhibition of 53BP1 is of interest for cancer therapy[48]. For example, silencing of the 53BP1 gene sensitizes glioma cells to ionizing radiation by prolonging cell cycle arrest and increasing apoptosis[49]. The expression of 53BP1 is increased in temozolomide-resistant glioblastoma cells and depletion of 53BP1 increases the potency of temozolomide against glioblastoma[50]. Potent and selective inhibitors of 53BP1 could therefore serve as lead compounds for drug development.

## Methods
### Protein purification
The tandem Tudor domain of 53BP1 (53BP1$^{TT}$, residues 1481–1603) was purified as previously described[12]. The W1495A, E1549P/D1550N (53BP1$^{TT}$-PN) and E1549C/E1567C (53BP1$^{TT}$-CC) mutants were similarly prepared. The p53K382me2 (residues 377–386) peptide was purchased from GenScript while the H4K$_C$20me2 (residues 12–25) was prepared from expression in *E. coli* with chemical installation of a

dimethyl-lysine analog as previously reported[51–53]. The two peptides were purified by reversed-phase chromatography using a preparative C18 column (Phenomenex).

### Preparation of small compounds
Synthesis of UNC2170, UNC2991, UNC3351, and UNC3474 and control ligands UNC1078 and UNC1118 was carried out by amide coupling reactions between the appropriately substituted benzoic acids and the appropriate amines using standard conditions as reported in detail in our prior publications[15,21]. Purification and characterization of these compounds followed the standard procedures in our prior publications[15,21].

### Nuclear magnetic resonance (NMR) spectroscopy
All NMR data were collected at 25 °C on a 700 MHz Bruker Avance III spectrometer equipped with a cryoprobe. Data were processed and analyzed with NMRPipe[54] and NMRView[55,56].

For the backbone resonance assignments of 53BP1$^{TT}$ in the presence of UNC3474, we prepared samples of 1 mM $^{15}$N- and $^{15}$N,$^{13}$C-labeled WT 53BP1$^{TT}$ with fivefold molar excess of non-labeled UNC3474 in 25 mM sodium phosphate, pH 7.0, 100 mM NaCl, 1.5 mM NaN$_3$, 0.3 mM DSS, 90% H$_2$O/10% D$_2$O. A series of standard NMR experiments including 2D $^1$H-$^{15}$N HSQC and $^1$H-$^{13}$C HSQC and 3D HNCACB, CBCA(CO)NH, HNCO, HN(CA)CO, HBHA(CO)NH, $^{15}$N-edited NOESY-HSQC and $^{13}$C-edited NOESY-HSQC were recorded[57].

To probe chemical shift perturbations, $^1$H-$^{15}$N HSQC spectra were acquired on the following samples: 0.2 mM $^{15}$N-labeled 53BP1$^{TT}$ WT and

53BP1$^{TT}$-PN in 25 mM sodium phosphate, pH 7.0, 1.5 mM NaN$_3$, 0.3 mM DSS, 90% H$_2$O/10% D$_2$O, free and incrementally titrated with various non-labeled inhibitors (UNC1078, UNC1118, UNC2170, UNC2991, UNC3474), as well as with H4K$_C$20me2 and p53K382me2 peptides. Chemical shift perturbations ($\Delta\delta$) were calculated using the Eq. (1):

$$\Delta\delta = \sqrt{(\Delta N/5)^2 + (\Delta H)^2} \tag{1}$$

where $\Delta N$ and $\Delta H$ are the corresponding differences in $^{15}N$ and $^1H$ chemical shifts between the bound and free forms of 53BP1.

The NMR relaxation measurements were carried out with 1 mM $^{15}N$-labeled 53BP1$^{TT}$, 53BP1$^{TT}$-PN, and 53BP1$^{TT}$ in the presence of saturating amounts of UNC3474 and UNC1118 in 25 mM sodium phosphate, pH 7.0, 100 mM NaCl, 1.5 mM NaN$_3$, 0.3 mM DSS, 90% H$_2$O/10% D$_2$O, using standard experiments[58,59]. The longitudinal and transverse $^{15}N$ relaxation rates R$_1$ and R$_2$ were determined using 12 and 14 relaxation delay times of 0.1–2 s and 4–120 ms, respectively. The first data point of each set of measurements was repeated for error assessment. The $^{15}N$-{$^1H$} nuclear Overhauser ratios were obtained from a reference experiment without proton irradiation and a steady-state experiment with proton irradiation for 3 s.

The protomer interconversion rates in the 53BP1$^{TT}$-UNC2970 complex were determined using 2D longitudinal $^1H$-$^{15}N$-heteronuclear ZZ-exchange NMR spectroscopy[22–24]. For these experiments, a series of 11 spectra were acquired with mixing times of 0, 100, 200, 300, 400, 500, 600, 700, 800, 900, and 1000 ms in 25 mM sodium phosphate, pH 7.0, 100 mM NaCl, 1.5 mM NaN$_3$, 0.3 mM DSS, 90% H$_2$O/10% D$_2$O. The interconversion rates were obtained from nonlinear fitting of auto peak and exchange peak intensities accounting for two-site exchange and $^{15}N$ R$_1$ relaxation using the equations previously described for a two-state model[24].

### X-ray crystallography

Crystals of 53BP1$^{TT}$-PN mutant (40 mg/mL) were obtained by the hanging drop vapor diffusion method, mixing 1 µL of the sample in 50 mM sodium phosphate, pH 7.5, 50 mM NaCl and 1 µL of the reservoir solution at 22 °C. Crystals of 53BP1$^{TT}$ WT (40 mg/mL) in complex with 14 mM of UNC3474, UNC3351, and UNC2991 molecules (all in 50 mM sodium phosphate, pH 7.5, 50 mM NaCl) and crystals of 53BP1$^{TT}$-CC mutant (15 mg/mL) in 50 mM Tris-HCl, pH 7.0, 100 mM NaCl, were obtained similarly. The reservoir solutions that produced crystals were: 1.5 M sodium/potassium phosphate, pH 6.0 (for free 53BP1$^{TT}$-PN); 2 M sodium formate, 0.1 M Bis-tris propane, pH 7.0 (for 53BP1$^{TT}$-UNC3474); 0.1 M sodium citrate tribasic, pH 5.6, 1 M ammonium phosphate monobasic (for 53BP1$^{TT}$-UNC3351); 1 M sodium formate, 0.1 M Bis-tris propane, pH 7.0 (for 53BP1$^{TT}$-UNC2991); and 0.1 M Bis-Tris, pH 6.5 (for 53BP1$^{TT}$-CC). Crystals of 53BP1$^{TT}$-CC were cryoprotected with 25% (w/v) xylitol. All other crystals were cryo-protected with 30% (v/v) glycerol.

Diffraction data for 53BP1$^{TT}$-PN and 53BP1$^{TT}$-UNC2991 were collected at the Cornel High Energy Synchrotron Source (beamline A1), while data for 53BP1$^{TT}$-UNC3474 (beamline 19-ID), 53BP1$^{TT}$-UNC3351 (beamline 19-ID), and 53BP1$^{TT}$-CC (beamline 19-BM) were collected at the Advanced Photon Source at Argonne National Laboratory. Diffraction data were processed with HKL2000. Phases were determined by molecular replacement using the crystal structure of 53BP1 tandem Tudor domain (PDB 2G3R[12]). Model building and refinement were carried out with COOT[60] and PHENIX[61]. All molecular representations were prepared with ChimeraX[62].

### Small angle X-ray scattering (SAXS)

SAXS measurements were performed at 10 °C and different concentrations (0.67, 1.33 and 2 mg/mL) of 53BP1$^{TT}$ WT, free and in complex with 10-fold molar excess of UNC3474 or UNC1118, in 25 mM sodium phosphate, pH 7.5, 15 mM NaCl, as well as on control samples

containing only the small molecule in the same buffer. SAXS data were collected at the Lawrence Berkeley National Laboratory SYBILS beamline 12.3.1 and analyzed using programs from the SAXS data analysis software ATSAS (version 2.4.2)[63,64] including PRIMUS, GNOM, AUTOPOROD, GASBOR and CRYSOL. PRIMUS (version 3.0) was used for initial data processing. Radii of gyration ($R_g$) were calculated using the Guinier approximation[65]. Distance distribution functions $P(r)$ and maximum particle dimensions were computed using GNOM. Molecular volumes and weights were estimated using AUTOPOROD. Ab initio reconstructions of molecular envelopes were done using GASBOR 2.3i. CRYSOL was used to back calculate the scattering curves and determine the goodness of fit.

### Isothermal titration calorimetry (ITC)

The ITC measurements were performed at 25 °C using a MicroCal iTC200 calorimeter. 53BP1$^{TT}$ and the small molecules (UNC2991, UNC3351, UNC3474) were in 50 mM Tris-HCl, pH 7.5, 50 mM NaCl. In a typical run, 53BP1$^{TT}$ was in the reaction cell at a concentration of 150 to 160 µM while the small molecule, in the titration syringe at a concentration of 0.8–2 mM, was delivered as 2 µL injections every 3 min. The titrations were paired with control experiments for heat of dilution. The titration of 53BP1$^{TT}$ with UNC1118 was performed using a MicroCal VP-ITC calorimeter. ITC data were fitted to a one-site model using a Levenberg-Marquardt non-linear regression algorithm programmed in the Origin 7.0 software.

### Sedimentation velocity analytical ultracentrifugation (AUC)

The AUC experiments were performed at 20 °C in 20 mM sodium phosphate, pH 7.5, with 50 mM NaCl. 53BP1$^{TT}$ was at a concentration of 20 µM. Measurements were recorded on an Optima analytical ultracentrifuge (Beckman Coulter) with an An-50 Ti analytical 8-place rotor using absorbance detection at 280 nm. Sample-filled AUC cells assembled with 12-mm charcoal-filled Epon centerpieces and quartz windows were subjected to 3 h of equilibration at 20 °C under vacuum prior to initiating the experiments at 50,000 rpm (corresponding to $182,000 \times g$ at the cell center and $201,600 \times g$ at the cell bottom) with continuous data acquisition for 20 h (200 scans) at 20 °C. The AUC data were analyzed with the c(s) model in SEDFIT[66] using a partial specific volume of 0.7335 mL/g.

### Cell biology

The foci-forming region (FFR; residues 1220–1711) of wild-type 53BP1 and mutants E1549P/D1550N (53BP1$^{FFR}$-PN) and E1549C/E1567C (53BP1$^{FFR}$-CC) were inserted into the retroviral vector POZ. The primers used to introduce the E1549P/D1550N mutations are: CACTGAAGT-GACGGCCCTCTCGCCCAATGAGTATTTCAGTGCAGGAGTGGTGAAAG G (forward) and CCTTTCACCACTCCTGCACTGAAATACTCATTGGG CGAGAGGGCCGTCACTTCAGTG (reverse). The primers used to introduce the E1549C/E1567C mutations are: TGCGATGAGTATTT-CAGTGCAGGAGTGGTGAAAGGACATAGGAAGGAGTCTGGGTGCCTGT ACTACAGCATTGAAAAAGAAGGCC (forward) and GCACCCAGACTC CTTCCTATGTCCTTTCACCACTCCTGCACTGAAATACTCATCGCACGA GAGGGCCGTCACTTCAG (reverse). The 53BP1$^{FFR}$ constructs were stably expressed at a moderate level in U2OS cells. To probe the effect of UNC3474 on 53BP1$^{FFR}$ IRIF formation, U2OS cells were grown on glass coverslips, pre-incubated for 1 h with various concentrations of UNC3474 and exposed to 1 Gy of X-ray radiation. After 1 h, the cells were fixed in 4% formaldehyde in PBS for 15 min at room temperature and blocked and permeabilized for 1 h in PBS containing 0.3% Triton X-100, 1% BSA, 10% fetal bovine serum. For the western blots, the antibodies used to detect 53BP1$^{FFR}$ and β-actin were monoclonal anti-Flag M2 produced in mouse (SigmaAldrich F1804, dilution 1:2000) and monoclonal anti-β-actin produced in mouse (Santa Cruz sc-47778, dilution 1:5000). For immunofluorescence, 53BP1$^{FFR}$ proteins (WT and 53BP1-PN) and γH2A.X were stained respectively with monoclonal anti-

HA-Tag (Cell Signaling Technology C29F4, dilution 1:1000) antibody produced in rabbit and monoclonal anti-γH2A.X (Ser139) (MilliporeSigma JBW301, dilution 1:1000) antibody produced in mouse. Secondary antibodies were goat anti-mouse IgG Alexa Fluor 594 conjugate (Thermo Fisher Scientific A-11005, dilution 1:2000) and goat anti-rabbit IgG Alexa Fluor 488 conjugate (Thermo Fisher Scientific A-11008, dilution 1:2000). Incubation with primary and secondary antibodies were done in PBS containing 1% BSA and 0.1% Triton X-100 for 1 h at room temperature. Coverslips were mounted using DAPI Fluoromount-G (Southern Biotech).

### Statistics and reproducibility
Cell samples were subjected to z-stack scanning with a total depth of 5 μm to comprehensively capture all 53BP1$^{FFR}$ and γH2A.X IRIF within the cell nuclei, which were fluorescently labeled with DAPI. The resulting 3D image datasets were subsequently transformed into 2D datasets using the maximum intensity projection technique in ImageJ (version 1.53a)[67]. To assess the quantification of 53BP1$^{FFR}$ and γH2A.X foci per nuclei, a customized CellProfiler (version 3.0)[68] pipeline was employed. During this analysis, the investigators conducting the study were kept blinded to ensure impartiality. No specific statistical method was employed to predetermine the sample size. For statistical analysis, to determine the $P$ values, a two-tailed $t$-test was applied using GraphPad Prism (version 9.0.0).

### Reporting summary
Further information on research design is available in the Nature Portfolio Reporting Summary linked to this article.

## Data availability
The crystallographic models and data have been deposited in the Protein Data Bank under accession codes "6MXX" (53BP1$^{TT}$-UNC2991), "6MXY" (53BP1$^{TT}$-UNC3351), "6MXZ" (53BP1$^{TT}$-UNC3474), "6MY0" (53BP1$^{TT}$-PN), and "8U4U" (53BP1$^{TT}$-CC). Source data are provided with this paper.

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

## Acknowledgements

We are very grateful to Nicolas Foloppe for insightful discussions and advice, to Kemin Tan at Argonne National Laboratory Advanced Photon Source (APS) for assistance with X-ray data collection and to Greg Hura at Lawrence Berkeley National Laboratory Advanced Light Source (ALS) for assistance with SAXS data collection. This research was supported by the US National Institutes of Health (NIH) awards R35 GM136262 and R01 CA132878 to G.M., US Department of Defense Ovarian Cancer Research Program awards W81XWH-16-1-0391/OC150206 and W81XWH-20-1-0322/OC190394 to M.V.B. and G.M., and NIH grants R01 CA208244 and R01 CA264900 and Gray Foundation Team Science Award to D.C. X-ray crystallography data were collected at the APS Structural Biology Center funded by the U.S. Department of Energy (DOE), Office of Biological and Environmental Research and operated for the DOE Office of Science by Argonne National Laboratory under con-tract DE-AC02-06CH11357. X-ray crystallography data were also col-lected at the Cornell High-Energy Synchrotron Source (CHESS), which is supported by the National Science Foundation under award DMR-1829070, and the Macromolecular Diffraction at CHESS (MacCHESS) facility, which is supported by NIH award P30 GM124166. The SAXS data were collected at the ALS SIBYLS beamline, supported by the DOE, Office of Biological Environmental Research, and by the NIH project ALS-ENABLE (P30 GM124169) and High-End Instrumentation Grant S10 OD018483.

## Author contributions

Conceptualization, G.M., G.C. and M.V.B.; Investigation, G.C., M.V.B., P.D., Q.H., B.B., J.R.T., D.J.S., A.D., M.T.P., L.I.J., and G.M.; Writing—original draft, G.M. and M.V.B.; Writing—review and editing, G.M., M.V.B., J.R.T., G.C., P.D., D.J.S., L.I.J., and S.V.F.; Supervision, G.M. and D.C.

## Competing interests

The authors declare no competing interests.
