## [Peer Review File · Nature Communications]

An autoinhibited state of 53BP1 revealed by small molecule antagonists and protein engineeringREVIEWER COMMENTS

Reviewer #1 (Remarks to the Author):

In this manuscript, Cui and colleagues study previously uncharacterized dimerization of the tandem Tudor domain (TTD) of 53BP1. The authors co-crystallized TTD with small molecule inhibitors and found that the inhibitors are sandwiched between two TTD protomers. This finding prompted the authors to fully investigate the ability of TTD to form dimers. Since dimerization blocks the binding site for the histone ligand (H4K20me2), the authors proposed that dimerization can provide the mechanism for auto-inhibition, and using a large set of approaches, including sophisticated NMR experiments, ITC, crystallography, SAXS, and mutagenesis (mutation to Cys is an excellent idea in particular) confirmed that TTD is capable of dimerization, which leads to autoinhibition of 53BP1 in cells.

Overall, this is a very well executed study that addresses an important biological question regarding the autoinhibition mechanism by which function of 53BP1 in the DNA damage response can be regulated. More often we focus on activation processes, however (auto)inhibition of PTM/histone binding proteins is by far less characterized, despite the fact it is as important as activation. Otherwise, processes will go on forever. This manuscript contains excellent quality data, conclusions are convincing and justified, and I enthusiastically support publication.

A minor advice, it will be of help to show a schematic of 53BP1 in Fig. 1 and to indicate where the H4K20me2 binding site is in Fig. 1a. Fig. 5b needs to be larger, currently, the green signal could be seen only at high magnification.

Reviewer #2 (Remarks to the Author):

53BP1 is an important reader protein of methylated histone and plays critical roles in DNA damage repair. This amazing manuscript identified a set of interesting chemical probes targeting 53BP1 to inhibit its histone binding ability. The authors previously found a 53BP1-binding antagonists, and found that the antagonists stabilize the auto-associated form of 53BP1 by making it locked in a closed state and unable to binding histone. They further utilized the utilized the NMR technique as the powerful tool to capture a lowly populated state of 53BP1, confirming that the existence of the auto-inhibited state in solution. The developed chemical probes reflect a new strategy in stabilizing the auto-inhibition form of a target protein rather than competing its ligand-binding site. The study would be of general interest to the readership of epigenetics, gene transcription, and chemical biology. The manuscript is well written and easy to follow with the figures clear and well organized. I only have very minor concerns for table1, and support its publication after they are corrected.

In table 1, for the listed space groups, "90.00", "120.00" should be "90", "120" respectively. The better to show Rwork and Rfree values in three decimal digits, for example, 0.210/0.230, etc. N/A (not applicable) could be used to replace some blanks in the table.

Reviewer #3 (Remarks to the Author):

In this manuscript, Cui et al. identified several antagonists of tandem Tudor domains of 53BP1 (53BP1TT) and elucidated the molecular basis of inhibition of these antagonists. By crystallographic, NMR, mutational, and biochemical analyses, they showed that all the antagonists were encapsulated in a cavity generated by two 53BP1TT molecules and the interaction between antagonists and 53BP1TT need dimerization of two 53BP1TT molecules. However, they did not show any direct evidences for the auto-associated form of 53BP1 for the TT domain, let alone a full-length protein. So, the saying of "pre-existing auto-associated form of 53BP1TT" is not fitful. Since the antagonists bind to the interface

of the dimer, it is expected that any mutation that destroys the dimer will prevent the antagonist from binding. Furthermore, a similar antagonist of 53BP1TT and similar three-dimensional (3D) arrangement have been reported by them previously (ACS Chem. Biol. 2015, 10, 4, 1072–1081). So, I am concerned that the significance and novelty of this manuscript are not significant enough to be published in Nature Communications.

I also have a couple of issues that can be considered for further improvement.

1. The reason of different binding affinities of these highly similar antagonists should be discussed in the manuscript, since it will guide the optimization of more effective antagonists for 53BP1TT.
2. Why the 53BP1TT-PN mutant was built? In another word, why a proline and an asparagine but not alanine (which is more common for the point mutation) were selected to replace E1549 and D1550?
3. Why the 53BP1TT-CC mutant was built on E1549 and E1567 but not on E1549-R1583, D1550-R1490, and D1550-K1505, since these residues also formed closed interactions.
4. Why “Importantly, 53BP1FFR-CC should be in a reduced state within mammalian cells”?
5. The molecular weight-marker should be shown in figure 5A and why the molecular weights of mutants are a little bigger than wildtype? The data of two mutants were not shown in figure 5B; Does this high concentration (50 or 100 μ M) of antagonist is toxic to cells? Furthermore, why the two mutants showed similar trend as wildtype in figure 5D?
6. The structural data and associated figure of 53BP1TT-CC should be omitted, since the resolution and quality of the data is too poor.

REVIEWER COMMENTS

Reviewer #1 (Remarks to the Author):

In this manuscript, Cui and colleagues study previously uncharacterized dimerization of the tandem Tudor domain (TTD) of 53BP1. The authors co-crystallized TTD with small molecule inhibitors and found that the inhibitors are sandwiched between two TTD protomers. This finding prompted the authors to fully investigate the ability of TTD to form dimers. Since dimerization blocks the binding site for the histone ligand (H4K20me2), the authors proposed that dimerization can provide the mechanism for auto-inhibition, and using a large set of approaches, including sophisticated NMR experiments, ITC, crystallography, SAXS, and mutagenesis (mutation to Cys is an excellent idea in particular) confirmed that TTD is capable of dimerization, which leads to autoinhibition of 53BP1 in cells.

Overall, this is a very well executed study that addresses an important biological question regarding the autoinhibition mechanism by which function of 53BP1 in the DNA damage response can be regulated. More often we focus on activation processes, however (auto)inhibition of PTM/histone binding proteins is by far less characterized, despite the fact it is as important as activation. Otherwise, processes will go on forever. This manuscript contains excellent quality data, conclusions are convincing and justified, and I enthusiastically support publication.

A minor advice, it will be of help to show a schematic of 53BP1 in Fig. 1 and to indicate where the H4K20me2 binding site is in Fig. 1a. Fig. 5b needs to be larger, currently, the green signal could be seen only at high magnification.

Response to the review:

We are very grateful to Reviewer #1 for their enthusiasm for our work and for the helpful suggestions. We have revised Figure 1 to include a schematic of 53BP1 in Figure 1a. We have also included a new panel (Figure 1d) in which the 53BP1^{TT}-H4K20me2 structure is now shown to highlight how the “encapsulated ligands” prevent the interaction of 53BP1^{TT} with H4K20me2. For comparison with 53BP1^{TT}-H4K20me2, the two binding interfaces in the 53BP1^{TT}-UNC3351 structure are shown in Figure 1d. In the revised manuscript, Figure 5b is larger and has been modified to show the cellular data for the two mutants. Finally, in the title of the manuscript, we replaced “chemical probes” by “small molecule antagonists” akin to “small molecule inhibitors” used by Reviewer #1.

Reviewer #2 (Remarks to the Author):

53BP1 is an important reader protein of methylated histone and plays critical roles in DNA damage repair. This amazing manuscript identified a set of interesting chemical probes targeting 53BP1 to inhibit its histone binding ability. The authors previously found 53BP1-binding antagonists and found that the antagonists stabilize the auto-associated form of 53BP1 by making it locked in a closed state and unable to binding histones. They further utilized the NMR technique as the powerful tool to capture a lowly populated state of 53BP1, confirming the existence of the auto-inhibited state in solution. The developed chemical probes reflect a new strategy in stabilizing the auto-inhibition form of a target protein rather than competing its ligand-binding site. The study would be of general interest to the readership of epigenetics, gene

transcription, and chemical biology. The manuscript is well written and easy to follow with the figures clear and well organized. I only have very minor concerns for table1 and support its publication after they are corrected.

In table 1, for the listed space groups, “90.00”, “120.00” should be “90”, “120” respectively. It is better to show Rwork and Rfree values in three decimal digits, for example, 0.210/0.230, etc. N/A (not applicable) could be used to replace some blanks in the table.

Response to the review:

We are very grateful to Reviewer #2 for the very positive evaluation of our manuscript. We concur with their suggestions. We have corrected Table 1 for the space group angles and the Rwork and Rfree values are now presented with three decimal digits. N/A was added where needed in Table 1.

Reviewer #3 (Remarks to the Author):

In this manuscript, Cui et al. identified several antagonists of tandem Tudor domains of 53BP1 (53BP1TT) and elucidated the molecular basis of inhibition of these antagonists. By crystallographic, NMR, mutational, and biochemical analyses, they showed that all the antagonists were encapsulated in a cavity generated by two 53BP1TT molecules and the interaction between antagonists and 53BP1TT need dimerization of two 53BP1TT molecules.

However, they did not show any direct evidences for the auto-associated form of 53BP1 for the TT domain, let alone a full-length protein. So, the saying of “pre-existing auto-associated form of 53BP1TT” is not fitful. Since the antagonists bind to the interface of the dimer, it is expected that any mutation that destroys the dimer will prevent the antagonist from binding.

Furthermore, a similar antagonist of 53BP1TT and similar three-dimensional (3D) arrangement have been reported by them previously (ACS Chem. Biol. 2015, 10, 4, 1072–1081). So, I am concerned that the significancy and novelty of this manuscript are not significant enough to be published in Nature Communications.

I also have a couple of issues that can be considered for further improvement.

1. The reason of different binding affinities of these highly similar antagonists should be discussed in the manuscript, since it will guild the optimization of more effective antagonists for 53BP1TT.

2. Why the 53BP1TT-PN mutant was built? In another word, why a proline and an asparagine but not alanine (which is more common for the point mutation) were selected to replace E1549 and D1550?

3. Why the 53BP1TT-CC mutant was built on E1549 and E1567 but not on E1549-R1583, D1550-R1490, and D1550-K1505, since these residues also formed closed interactions.

4. Why “Importantly, 53BP1FFR-CC should be in a reduced state within mammalian cells”?

5. The molecular weight-marker should be shown in figure 5A and why the molecular weights of mutants are a little bigger than wildtype? The data of two mutants were not shown in figure 5B; Does this high concentration (50 or 100 μ M) of antagonist is toxic to cells? Furthermore, why the two mutants showed similar trend as wildtype in figure 5D?

6. The structural data and associated figure of 53BP1TT-CC should be omitted, since the resolution and quality of the data is too poor.

Response to the review:

We are very grateful for the thoughtful comments provided by Reviewer #3. We have tried our best to address the different issues that were raised. As a result, we believe that our revised manuscript is much improved.

Comment on “However, they did not show any direct evidence for the auto-associated form of 53BP1 for the TT domain, let alone a full-length protein. So, the saying of “pre-existing auto-associated form of 53BP1TT” is not fitful.”

We thank Reviewer #3 for this comment, allowing us to provide further clarification. We believe that our study demonstrates the existence of a pre-existing auto-associated form of 53BP1^{TT}. In this homodimeric form of 53BP1^{TT}, the H4K20me2 and p53K382me2 binding surface is buried. Therefore, this 53BP1^{TT} dimer is “autoinhibited” for chromatin binding. We notably designed mutations that prevent the auto-association of 53BP1^{TT}. We showed that these mutations do not affect the histone and p53 binding surface of 53BP1^{TT}. Mutated 53BP1^{TT} still binds H4K20me2 and p53K382me2, as well as the larger small molecule antagonists that target the methyllysine binding cage of 53BP1^{TT}. However, the “encapsulated antagonists” do not bind mutated 53BP1^{TT} that cannot homodimerize. To us, this observation is clear evidence, albeit indirect, for a pre-existing auto-associated form of 53BP1.

By introducing two cysteines in 53BP1^{TT}-CC, we were also able to stabilize the auto-associated form of 53BP1 and showed using X-ray crystallography that 53BP1^{TT}-CC adopts the expected structure for an auto-inhibited dimer capable of binding the “encapsulated antagonists.”

We believe that our extensive analysis *in vitro* provided a rigorous framework for performing experiments in cells. We showed that mutations in 53BP1^{TT} that prevent its dimerization and

prevent binding to the “encapsulated antagonists” also render 53BP1^{FFR} insensitive to these ligands in cells. Wild type 53BP1^{FFR} is inhibited by the “encapsulated antagonists.” Therefore, we conclude that 53BP1^{TT} can auto-associate in an auto-inhibited state in the context of 53BP1^{FFR} in cells.

Comment on “Since the antagonists bind to the interface of the dimer, it is expected that any mutation that destroys the dimer will prevent the antagonist from binding.”

We understand the point made by Reviewer #3. To clarify; the antagonists bind the methyllysine binding cage of 53BP1^{TT} as can be seen in the crystal structures. Based on the crystal structures, it is not possible to infer that the antagonists bind a dimer in solution. These antagonists were identified in a screen selecting ligands that can displace H4K20me2 from 53BP1^{TT}. 53BP1^{TT} is a monomer when bound to H4K20me2 as shown using NMR spectroscopy and X-ray crystallography. We had initially assumed that the observed dimerization in the first crystal structure of 53BP1^{TT} bound to an “encapsulated antagonist” was simply the result of crystal packing since 53BP1^{TT} had only been shown to be a monomer in previous solution studies. However, seeing a similar dimer in several crystal structures led us to test the hypothesis that a lowly populated dimer exists in solution and that the “encapsulated antagonists” only bind this pre-existing autoinhibited dimer and shift the equilibrium towards this dimer. We selected the 53BP1^{TT} mutations in such a way that they would be minimal and disrupt dimerization but would not involve any of the 53BP1^{TT} residues that contact the “encapsulated ligands.” Our results showed that these mutations precluded the “encapsulated antagonists” from interacting with 53BP1^{TT}. Our interpretation of these results is that the “encapsulated antagonists” can only bind a 53BP1^{TT} dimer, therefore demonstrating the pre-existence of such a dimer. The 53BP1^{TT} dimer is a lowly populated “invisible state” of 53BP1^{TT} that is revealed by these “encapsulated antagonists.”

One could have argued that an “encapsulated antagonist” glues two 53BP1^{TT} together in a dimer. This is not the case. A pre-existing dimer must be present for the antagonist to bind.

Comment on “Furthermore, a similar antagonist of 53BP1^{TT} and similar three-dimensional (3D) arrangement have been reported by them previously (ACS Chem. Biol. 2015, 10, 4, 1072–1081).”

We fully appreciate Reviewer #3’s comment. We would like to further clarify that this previous study referenced in our manuscript was the starting point for our current work. In this previous study, we identified and characterized a small molecule antagonist that displaces H4K20me2 from 53BP1^{TT}. We had not considered the possibility that the dimer observed in the crystal was not a crystallization artifact. Furthermore, we had not considered the concept of inhibition of 53BP1 chromatin recruitment by ligand-mediated stabilization of the dimer. Also, we had not shown that an “encapsulated antagonist” could block the formation of 53BP1 foci.

1. The reason of different binding affinities of these highly similar antagonists should be discussed in the manuscript, since it will guide the optimization of more effective antagonists for 53BP1^{TT}.

Indeed, obtaining more effective antagonists is our goal. The differences in affinity are small, making it difficult to provide a clear explanation by examining the structures. However, one can see that when comparing the lowest and higher affinity compounds (UNC2170 and UNC3474), the isopropyl group in UNC3474, larger than the bromine in UNC2170, fits better in a cavity of 53BP1^{TT}. In the manuscript, our focus was to use the compounds as chemical probes to test the hypothesis of an auto-associated form of 53BP1 that blocks the histone binding surface of 53BP1 tandem Tudor domain. In future studies, guided by the structures, new compounds carrying varied substitutions at different sites will be tested.

2. Why the 53BP1^{TT}-PN mutant was built? In another word, why a proline and an asparagine but not alanine (which is more common for the point mutation) were selected to replace E1549 and D1550?

We selected a proline and asparagine by rational design to minimize any effects of the mutations on the structure of 53BP1^{TT} while preventing the formation of six inter-protomer salt bridges to impede dimerization. The Pro-Asn pair stabilizes a turn in a loop of 53BP1^{TT}. By X-ray crystallography, we could show that the 53BP1^{TT}-PN mutant closely resembles wild type 53BP1^{TT}. We showed that 53BP1^{TT}-PN still bound H4K20me2 and p53K382me2 like wild type 53BP1^{TT} *in vitro*. We also showed that 53BP1^{FFR}-PN could form foci in cells like wild type 53BP1^{FFR}.

3. Why the 53BP1^{TT}-CC mutant was built on E1549 and E1567 but not on E1549-R1583, D1550-R1490, and D1550-K1505, since these residues also formed closed interactions.

These mutations were selected after analyzing the highest resolution 3D structures reported in our manuscript (PDB entry 6MXY in Table 1). We concluded that residues E1549 and E1567 were best positioned for engineering the disulfide bridges with minimal disruption to the 53BP1^{TT} structure. To avoid biases, we have now also used a software (Disulfide by Design 2.0;

referenced in the manuscript) that selects the most optimal residues for engineering disulfide bridges in proteins. Noticeably, when using the highest resolution structure in Table 1 (PDB entry 6MXV), the only option generated for inter-molecular disulfide bridges was E1549/E1567.

Importantly, the formation of these two disulfide bridges is a test of the pre-existing dimer hypothesis. As shown in our manuscript, disulfide bridges did form between E1549 and E1567, as expected, and preserved the relative orientations of the two 53BP1^{TT} protomers in the dimer. 53BP1^{TT}-CC has the same dimeric organization as 53BP1^{TT} bound to the “encapsulated antagonists.” Therefore, with the disulfide bridges we were able to trap or stabilize a pre-existing dimer.

4. Why “Importantly, 53BP1^{FFR}-CC should be in a reduced state within mammalian cells”?

Indeed, this comment needs clarification. We have modified the text to point out that 53BP1^{FFR}-CC is expected to be in a reduced state in mammalian cells owing to the reductive nuclear environment and noted that antioxidant enzymes are activated in response to ionizing radiation. We have included three references related to the reductive nuclear environment.

5. The molecular weight-marker should be shown in figure 5A and why the molecular weights of mutants are a little bigger than wildtype? The data of two mutants were not shown in figure 5B; Does this high concentration (50 or 100 μ M) of antagonist is toxic to cells? Furthermore, why the two mutants showed similar trend as wildtype in figure 5D?

The molecular weight maker has now been included in Figure 5. The gel shown in Figure 5a was slightly tilted. In the corrected figure, only the PN mutant (D1549P/D1550N) appears slightly bigger than the wild type protein. This difference in protein migration in SDS-PAGE might be caused by the change in charge of the “PN loop,” from negative to neutral. However, we cannot provide a conclusive explanation.

We have now updated Figure 5b to show the cellular data for the two mutants. We determined that the high concentration of antagonists does cause cellular toxicity that becomes apparent after about 72 hours of treatment with the antagonists. The cells were fixed for data analysis two hours post treatment with the antagonists as indicated in the Methods.

Regarding Figure 5d, we have clarified the text in response to the Reviewer #3’s comment. In Figure 5d, the γ H2AX foci are reporters for DNA double-strand breaks in cells. In this experiment, we should not detect any differences between the cells expressing wild type or mutated 53BP1.

6. The structural data and associated figure of 53BP1^{TT}-CC should be omitted, since the resolution and quality of the data is too poor.

We acknowledge the Reviewer #3’s comment. We made it clear in the revised manuscript that we derived a model based on low resolution X-ray diffraction data. We opted to keep these data and figure because the low-resolution structure is sufficient to demonstrate that the dimeric arrangement is preserved in 53BP1^{TT}-CC. The data have been deposited in the Protein Data

Bank. Reviewer #1 also noted that this particular aspect of our work using cysteines to form disulfide bridges was an excellent idea.

REVIEWERS' COMMENTS

Reviewer #1 (Remarks to the Author):

The authors have adequately addressed my previous comments.

Reviewer #2 (Remarks to the Author):

The authors have fully addressed my previous concerns and I suggest that it be accepted and published in a timely manner.

Reviewer #3 (Remarks to the Author):

Thanks for the author point by point response. Since all the evidence for homodimer is indirect, I persist in the saying of "pre-existing auto-associated form of 53BP1TT" is not fitful. And the data quality of 53BP1TT-CC is not acceptable, the higher resolution data of this mutant should be provided. By the way, I agree with reviewer #1 that the disulfide bridges of two cystines is an excellent idea, but that does not mean a crystal structure with a resolution of 3.79 Ångstrom acceptable. And it is a pity that 53BP1TT-CC could not lock the 53BP1TT in a dimerization status in cell to block the recruitment of 53BP1TT to the chromatin to support the main idea "auto-associated form of 53BP1 — autoinhibited for chromatin binding".

REVIEWER COMMENTS

Reviewer #1 (Remarks to the Author):

The authors have adequately addressed my previous comments.

Reviewer #2 (Remarks to the Author):

The authors have fully addressed my previous concerns and I suggest that it be accepted and published in a timely manner.

Response to the review:

We are very grateful to Reviewer #1 and Reviewer #2 for their enthusiasm for our work. Because of their thoughtful suggestions in the previous review, we believe that our manuscript is greatly improved.

Reviewer #3 (Remarks to the Author):

Thanks for the author point by point response. Since all the evidence for homodimer is indirect, I persist in the saying of “pre-existing auto-associated form of 53BP1^{TT}” is not fitful. And the data quality of 53BP1^{TT}-CC is not acceptable, the higher resolution data of this mutant should be provided.

By the way, I agree with reviewer #1 that the disulfide bridges of two cystines is an excellent idea, but that does not mean a crystal structure with a resolution of 3.79 Ångstrom acceptable. And it is a pity that 53BP1^{TT}-CC could not lock the 53BP1^{TT} in a dimerization status in cell to block the recruitment of 53BP1^{TT} to the chromatin to support the main idea “auto-associated form of 53BP1 — autoinhibited for chromatin binding”.

Response to the review:

We thank Reviewer #3 for providing this review. In response to the comments of Reviewer #3, we have reprocessed our crystallography data and were able to significantly improve the coordinate model for 53BP1^{TT}-CC. The new data have been deposited in the Protein Data Bank under accession code 8U4U. Table 1 has been updated.

We respectfully disagree with Reviewer #3 regarding the removal of the 53BP1^{TT}-CC structure from our manuscript. We have chosen to include this structure. Although the resolution is low at 3.79 Å, the data clearly demonstrate that the two disulfide bridges form as expected and that the autoinhibitory dimeric arrangement identified in the 53BP1^{TT}-UNC3474 complex is preserved in 53BP1^{TT}-CC .

The image shows the 2Fo-Fc map contoured at 1σ level and the Fo-Fc map contoured at 3σ level for one of the disulfide bridges between chains A and B.